

# The importance of individuals of different sizes in the population maintenance of a palm species used by the Fulni-ô Indigenous People in northeast Brazil

Juliana Loureiro Almeida Campos[1,2], Elcida de Lima Araújo[3], Aldicir Scariot[4], Eduardo Teles Barbosa Mendes[5], Rita de Cássia Quitete Portela[5] and Ulysses Paulino Albuquerque[6]

[1] Instituto de Biodiversidade e Sustentabilidade, Universidade Federal do Rio de Janeiro, Macaé, Rio de Janeiro, Brazil

[2] Departamento de Biologia, Universidade Federal Rural de Pernambuco, Recife, Pernambuco, Brazil

[3] Departamento de Botânica, Universidade Federal de Pernambuco, Recife, Pernambuco, Brazil

[4] Laboratório de Ecologia e Conservação, Embrapa Recursos Genéticos e Biotecnologia, Brasília, Distrito Federal, Brazil

[5] Departamento de Ecologia, Instituto de Biologia, Universidade Federal do Rio de Janeiro, Rio de Janeiro, Brazil

[6] Laboratório de Ecologia e Evolução de Sistemas Socioecológicos (LEA), Departamento de Botânica, Universidade Federal de Pernambuco, Recife, Pernambuco, Brazil

Corresponding author
Juliana Loureiro Almeida Campos, loureiroju@hotmail.com

## ABSTRACT

Factors such as climate, soil characteristics, habitat type, and land management practices can influence the demography of plant populations harvested by Indigenous Peoples and local communities. Here, we assessed the demographic responses of the palm *Syagrus coronata* to varying leaf harvest frequencies by the Fulni-ô Indigenous People in sites with different environmental and anthropogenic conditions in Águas Belas, Pernambuco, northeast Brazil. The leaves of this species are primarily harvested for handicraft production. In collaboration with local artisans, we conducted a participatory workshop where they identified harvest locations on a regional map. Plots and subplots were established in six of these sites, and the total height of all *S. coronata* individuals was recorded. We monitored survival and growth over three consecutive years and counted infructescences on reproductive individuals every three months during the first two years. Newly recruited individuals were also recorded and measured. Environmental variables (light availability, air temperature, and humidity) were measured quarterly in the first year, and soil samples were collected for chemical and physical analysis. We performed a principal component analysis (PCA) to evaluate differences among sites based on environmental and anthropogenic variables. Using demographic data, we constructed integral projection models (IPMs) and conducted a life table response experiment (LTRE) analysis to estimate vital rates and deterministic population growth rates ($\lambda$) for each population and sampling interval. Our results showed that *S. coronata* populations under high harvest frequencies declined during the study period. In contrast, populations with lower harvest frequencies were more influenced by the growth of smaller individuals, though seedling recruitment was reduced, highlighting the need to preserve these younger plants. Higher air temperatures,
nutrient availability, and soil pH likely contributed to low adult fecundity and reduced recruitment. Additionally, cattle and livestock presence may have further hindered recruitment by trampling and grazing on smaller plants. In populations subjected to intermediate and high harvest frequencies, larger individuals had the greatest impact on population growth rates. However, these individuals exhibited lower survival, suggesting that harvest pressure may negatively affect this vital rate, as the Fulni-ô harvesters preferentially target larger juveniles and adults for their more substantial leaves. Based on these findings, we recommend management strategies to support *S. coronata* conservation while ensuring sustainable harvesting and safeguarding Fulni-ô handicraft production.

## INTRODUCTION

Non-timber forest products (NTFPs) are widely collected by human populations worldwide and play a crucial role in supporting the subsistence, autonomy, and income generation of local communities (*Matias et al., 2018*; *Alcântara, Lucena & Cruz, 2022*). It is estimated that approximately 50,000 wild species are used by human populations across the globe, with seventy percent of the most vulnerable communities depending on these species as a nutritional resource (*IPBES, 2022*). The harvesting of entire plants or specific plant parts can have significant consequences for their populations, including changes in vital rates such as survival, growth, and reproduction (*Martínez-Ramos, Anten & Ackerly, 2009*; *Navarro, Galeano & Bernal, 2011*; *Lopez-Toledo et al., 2012*), as well as alterations in resource allocation between vegetative and reproductive structures (*Endress, Gorchov & Berry, 2006*; *Gaoue & Ticktin, 2008*). These effects vary considerably depending on factors such as the species' life history, the regenerative potential of the harvested part, the environmental characteristics of the habitat, the cultural value of the species, and the frequency of harvesting (*Ticktin, 2004*).

Leaves represent a renewable resource for plants, and although their harvest rarely leads to the death of the individual, it can negatively affect long-term plant performance. Documented impacts include reduced production of reproductive structures (*Mandle, Ticktin & Zuidema, 2015*; *Lopez-Toledo et al., 2018*), increased allocation of resources to the production of new leaves (*Duarte & Montúfar, 2012*; *Lopez-Toledo et al., 2018*), and even a decrease in the rate of new leaf production (*Mendoza, Piñero & Sarukhan, 1987*). The effects of leaf removal have also been observed on vital rates, including increased mortality (*Endress, Gorchov & Noble, 2004*) and reduced population growth rates (*Endress, Gorchov & Berry, 2006*; *Valverde, Hernandez-Apolinar & Mendoza-Amaro, 2006*). However, studies on survival rates have reported divergent findings, with some showing no significant negative impacts (*Zuidema, De Kroon & Werger, 2007*; *Martínez-Ramos, Anten & Ackerly, 2009*) and others demonstrating clear negative effects following leaf removal (*Endress, Gorchov & Berry, 2006*; *Hernández-Barrios et al., 2012*; *Lopez-Toledo et al., 2012*).

Until recently, most studies evaluating the effects of NTFP harvest overlooked other variables that could influence the demographic behavior of harvested populations. Factors such as different land uses, management practices (*e.g.*, fire), habitat characteristics (including soil type, rainfall patterns, light availability, and topography), and more recently, climate change, can all significantly affect the population dynamics of harvested species (*Martínez-Ramos, Anten & Ackerly, 2009*; *Mandle & Ticktin, 2012*; *Baldauf et al., 2015*; *Sá, Scariot & Ferreira, 2020*; *Gaoue et al., 2019*; *Hart-Fredeluces & Ticktin, 2019*). For example, some species demonstrate greater resistance to harvest pressure in certain environmental conditions (*Baldauf et al., 2015*), even when fire is used as a management tool (*Hart-Fredeluces & Ticktin, 2019*). Climate change appears to interact with harvest effects, potentially altering the demographic responses of certain species (*Martínez-Ramos, Anten & Ackerly, 2009*; *Gaoue et al., 2019*; *Torres-Garcia et al., 2020*). These findings suggest the existence of complex interactions between habitat type, climatic conditions, and harvest intensity in determining population growth rates of species harvested by humans.

Consequently, fluctuations in populations of harvested species cannot be attributed solely to harvesting practices. Studies that fail to account for habitat characteristics, management approaches, and harvesting rates may draw incomplete or misleading conclusions about the sustainability of harvest activities. True sustainability in extractive practices occurs when the population's renewal rate compensates for the impact of harvesting, as indicated by a population growth rate ($\lambda$) equal to or greater than 1, which demonstrates that the population remains viable and capable of growth (*Freckleton et al., 2003*).

Palm trees represent one of the most important plant groups utilized by local communities worldwide, often forming an integral part of their livelihoods (*Lopez-Toledo et al., 2018*; *Abdullah et al., 2020*; *Andrade-Erazo et al., 2020*). In Brazil, the Fulni-ô Indigenous People maintain a particularly strong relationship with the palm species *Syagrus coronata* (Mart.) Becc., locally known as "Ouricuri" (*Campos et al., 2019*). This species plays a fundamental role in the culture, economy, and daily life of many people in Brazil's semi-arid region. For the Fulni-ô, this palm is deeply embedded in their routine activities, religious practices, and handicraft production, being used to create items such as rugs, mats, bags, hats, and clothing from its mature leaves. Historically, the trade of these handicraft products represented the primary source of income for the Fulni-ô while the leaves were also traditionally used in the construction of houses in Indigenous villages until the mid-1930s (*Pinto, 1956*).

The harvesting of NTFP has significant potential to affect the demographic structure of plant populations and consequently influence the sustainability of these extractive practices (*Ticktin, 2004*). In this study, we investigate the demographic responses of *S. coronata* palm populations to varying frequencies of leaf harvesting by the Fulni-ô Indigenous People across sites with different environmental and anthropogenic characteristics. Based on our findings, we aim to develop specific recommendations for leaf harvesting and management practices that will contribute to the conservation of *S. coronata* populations while supporting the sustainability of the traditional harvesting activities carried out by Indigenous Peoples.

## MATERIAL AND METHODS

### Study area

We conducted data collection in the municipality of Águas Belas (9°07′03″S, 37°07′06″W), located 311 km from Recife, the capital of Pernambuco state in northeastern Brazil, within the Caatinga biome. The vegetation is characteristic of the Caatinga, dominated by xerophytic, deciduous, and thorny species (*Araújo, Castro & Albuquerque, 2007*). The regional climate features two distinct seasons: a prolonged dry season lasting 5 to 9 months and a brief rainy season occurring between May and July (*Prado, 2003*). Águas Belas has a semi-arid climate (BShw' according to the *Köppen, 1948*), with an average annual temperature of 23.6 °C and mean annual precipitation of 621 mm (*APAC, 2024*). The predominant soil types are Regosols and Leptosols (*IUSS Working Group WRB, 2022*). The region is characterized by granitic mountain ranges and Cretaceous-origin plateaus (*Andrade-Lima, 1960*). Notably, these plateaus contain high-altitude swamps locally known as "hill marshes"—a unique vegetation physiognomy within the Caatinga biome. These areas support deciduous to sub-deciduous forests transitioning to sub-evergreen forests with evergreen species, exhibiting milder environmental conditions compared to surrounding vegetation types (*Rodrigues et al., 2008*). It is in these specific swamp environments that the Fulni-ô harvest leaves from *S. coronata* palms, which serve as raw material for their traditional handicraft production.

### The Fulni-ô Indigenous People

The Fulni-ô are one of seven Indigenous groups in Pernambuco state (ISA, 2007). Despite close proximity to non-indigenous settlements, they have preserved their cultural traditions. The Fulni-ô maintain their native Yathê language, the musical Toré tradition, and a secret religious ritual called "Ouricuri" practiced from September to December (*Silveira, Marques & Silva, 2012*). The Fulni-ô inhabit a 11,500-hectare Indigenous territory located 500 m from Águas Belas city, with two primary villages spaced 4 km apart. The main village has approximately 3,430 inhabitants, while the more rural Xixiakhlá village has about 100 residents. During their religious rituals, some community members temporarily inhabit a third village—which shares the same name as the ritual, Ouricuri—situated 6 km from the main settlement.

Economically, the Fulni-ô rely on handicraft production and sales, musical performances in other municipalities, tertiary sector employment (*Campos, 2011*), and land leasing (*Campos et al., 2018*). While agriculture, hunting and fishing occur occasionally, these activities primarily serve family subsistence. Their handicrafts incorporate wood and seeds from native Caatinga species (deciduous/xerophytic vegetation) surrounding their villages, along with leaves from *S. coronata* palms. These palms grow in high-altitude swamps—called "serras" by the Fulni-ô—located within or near their territory. Currently, many artisans (particularly elders) obtain palm leaves through purchase from younger community members or non-Fulni-ô suppliers. The leaves are sold weekly in the village or at Águas Belas' local market.

## The Ouricuri palm, *Syagrus coronata*

*S. coronata* occurs in the states of Pernambuco, Alagoas, Sergipe, Bahia, and northern Minas Gerais, growing in Caatinga vegetation, remnants of semi-deciduous forests, and transition zones to restinga (Atlantic Forest) and Cerrado (*Lorenzi, 2010*). This palm species has a solitary, erect stipe reaching 3–10 m in height and 15–25 cm in diameter, covered with leaf scars. As a monoecious species, it produces both male and female flowers on the same inflorescence. The fruits are yellowish with a brownish tomentum, measuring 2.5–3.0 cm long, with a sweet-tasting mesocarp (*Lorenzi, 2010*). While the palm fruits throughout the year, fruit production peaks between May and August, with ripening occurring from October to December (*Lorenzi, 2010*).

The Ouricuri palm is widely used by Indigenous Peoples and local communities inhabiting the Caatinga region. Both fruit pulp and seeds are edible and can be consumed raw or cooked. They are also processed into ouricuri flour for making cakes, sweets, cookies, and bread (*Crepaldi, Salatino & Rios, 2004*). Additionally, the seeds yield an oil of high nutritional value used in cooking and soap production. The water from immature seeds, known as ouricuri milk, serves as a key ingredient in regional dishes such as "umbuzadas" (a traditional food from northeastern Brazil) and various fish preparations (*de Lima Rufino et al., 2008*). The leaves are used for crafting handmade items and as animal fodder (*de Lima Rufino et al., 2008*; *Andrade et al., 2015*). Medicinally, the water from immature seeds is used to treat eye infections, mycoses, and wounds (*de Lima Rufino et al., 2008*). Beyond its importance to local communities, *S. coronata* serves as a crucial food resource for the endangered Lear's macaw (*Anodorhynchus leari* Bonaparte) (*Andrade et al., 2015*; *IUCN, 2019*), a bird species endemic to the "Raso da Catarina" region and Serra Branca Environmental Protection Area in Bahia.

For our study, we classified individuals as seedlings when they were identified in subplots during the first year of data collection with one or more entire, lanceolate, and narrow eophylls, including those that appeared in subsequent time intervals (recruits). We considered young individuals as those with fully segmented leaf blades and aerial stipes (either visible or covered by leaf sheaths) but showing no signs of reproduction. Reproductive individuals were identified by their fully segmented leaf blades, aerial stipes, and presence of reproductive structures. We further categorized all individuals into three height classes: small ($\leq 2.5$ m), intermediate (2.6–5 m), and large (>5 m). This categorization was implemented solely to illustrate the size distribution of individuals referenced in our results and discussion. During fieldwork, we recorded reproductive individuals as short as 0.9 m in height. For their handicraft production, the Fulni-ô primarily harvest leaves from intermediate and large *S. coronata* individuals.

## Legal aspects

The research project was approved by the National Commission of Ethics in Research (CAAE 24211014.0.0000.5207), the National Indian Foundation (Authorization No. 04/AAEP/PRES/2015), the National Historical and Artistic Heritage Institute (Case No. 2000.000203/2014-35), and the System of Authorization and Information in Biodiversity (Authorization No. 41944-1). These institutions are responsible for approving research

involving Indigenous Peoples and local communities in Brazil. In accordance with Fulni-ô cultural protocols, the community leaders provided verbal consent after reviewing and approving the research, which was accepted by all regulatory bodies given the Indigenous context.

### Identification of harvest sites and characterization of harvest regimes of *S. coronata* leaves

We initially identified members of the Fulni-ô Indigenous People who utilize *S. coronata* leaves for handicraft production using the snowball sampling technique. This method involves identifying key informants who then recommend other experts, eventually including all community specialists in the research process (*Albuquerque et al., 2014*). Subsequently, we conducted a participatory workshop (*Campos et al., 2018*) where participants marked all leaf harvest locations on a regional map and provided information about harvesting frequency at each site. From these locations, we selected six sites representing different harvest frequencies: two with low frequency (harvested approximately once monthly per harvester), two with intermediate frequency (harvested twice monthly), and two with high frequency (harvested weekly). The absence of completely undisturbed control areas in the Águas Belas region, as confirmed by workshop participants, prevented the inclusion of a no-harvest control site.

### Effects of leaf harvest on population dynamics of *S. coronata*

At each of the six *S. coronata* populations (Fig. 1), we established two permanent 50 × 50 m plots (minimum 100 m apart), totaling 1.5 ha. During initial plot establishment (July 2014, T0), we measured the total height (from ground to crown apex using graduated rods or tape measures) and tagged all non-seedling individuals with numbered plates. Over three consecutive years (June 2015/T1, June 2016/T2, July 2017/T3), we monitored survival, height growth, and recruitment of new individuals.

For seedling assessment, we established five 10 × 10 m subplots per main plot (four corners and center) at T0. All subplot seedlings were tagged and measured initially, with subsequent monitoring of survival, growth, and new recruitment during annual resampling. Plants that were missing were recorded as mortalities. Additionally, we counted infructescences on reproductive individuals quarterly during the first two study years.

### Additional information about *S. coronata* leaf harvesting regimes

We conducted semi-structured interviews (*Albuquerque et al., 2014*) with harvesters to obtain additional information about harvesting regimes and the characteristics that make certain individuals preferable for harvesting. Additionally, we counted the number of new leaves on each *S. coronata* individual every three months during the first 12 months of the study (between July 2014 and June 2015). For this procedure, we marked the newest leaf of each individual with waterproof red ink in July 2014 and again during each quarterly visit. We also recorded the total number of leaves per individual in both July 2014 and June 2015. We summed the number of new leaves produced annually by each palm, then
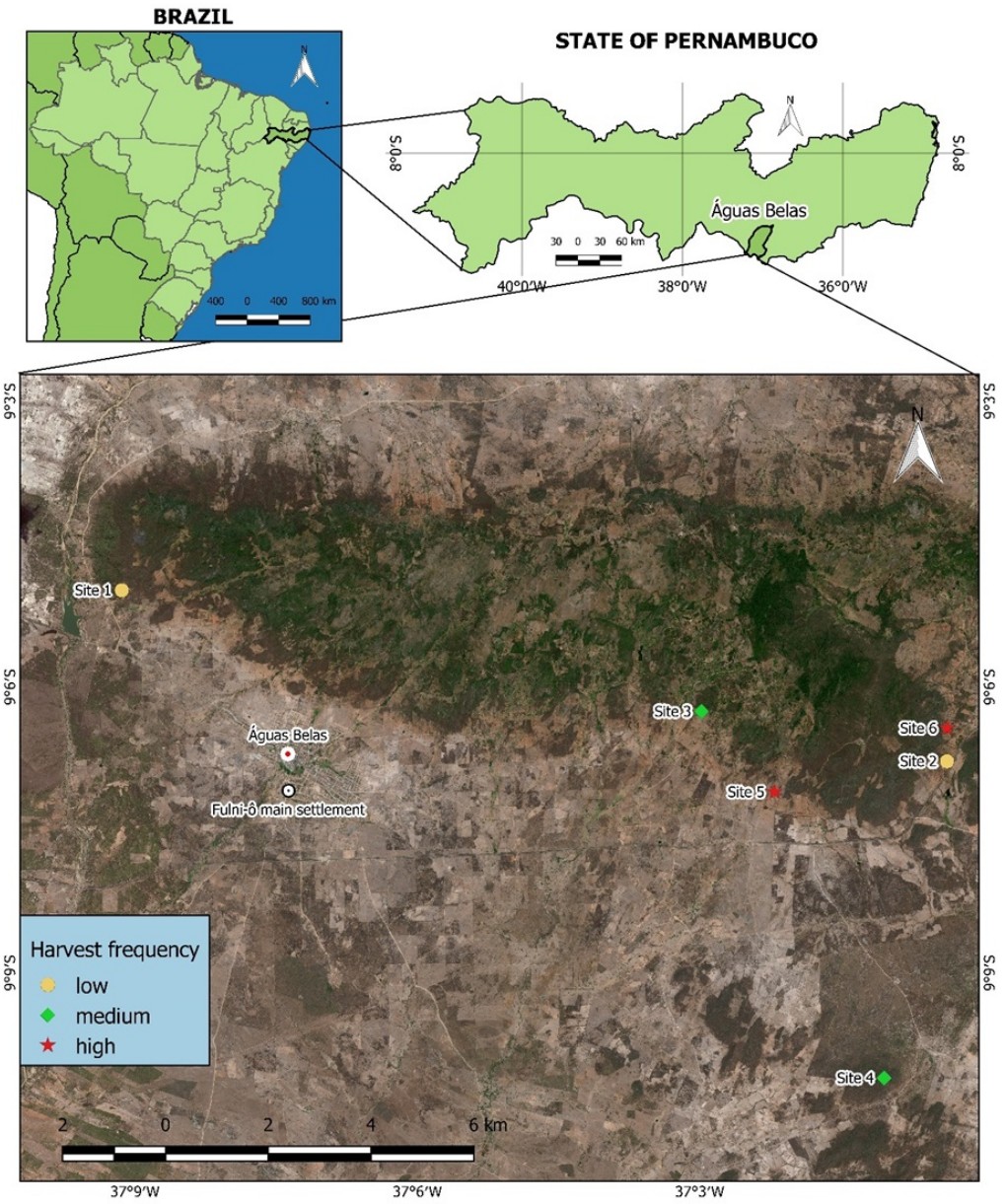

**Figure 1** Harvest sites for *S. coronata* leaves identified by Fulni-ô artisans in Águas Belas, Pernambuco, northeast Brazil. Site 1 and Site 2: low harvest frequency; Site 3 and Site 4: intermediate harvest frequency; Site 5 and Site 6: high harvest frequency. Created using QGIS (*QGIS Development Team, 2024*).

calculated the annual number of harvested leaves for each *S. coronata* individual using the following formula:

Annual harvested leaves = (Total leaves July 2014−Total leaves June 2015) + Annual number of new leaves.

## Characterization of the sampling sites

Luminosity, air temperature and humidity in each plot were recorded quarterly between June 2015 and June 2016 using a thermohygrometer. These environmental variables were measured at the four corners and the center of each permanent plot between 10 am and 12 pm, totaling four measures. We also sample soil at a depth of 20 cm at three points in each plot, and subsequently mixed them and sent to the Laboratory of Environmental Soil Chemistry of the Federal Rural University of Pernambuco for chemical and physical composition analysis. We assessed soil pH, Ca+, Mg+, Al+, Na+, K+, P+, H+Al, organic carbon (OC) and organic matter (OM) content. These variables were included due to differences in slopes and other landscape characteristics among the study areas, which led us to question whether they could also influence the demographic responses of the population. The characteristics of sampling sites are depicted in Table 1 and in Fig. 2.

## Data analysis

We calculated the average number of leaves harvested annually from each *S. coronata* individual per harvest frequency by summing the annual harvested leaves of all individuals within the same harvest frequency category and dividing by the total number of individuals in that frequency category (Table S1).

We performed a principal component analysis (PCA) (*Legendre & Legendre, 2012*) to assess differences among sites based on environmental and anthropogenic predictors. The variables included in the PCA were: soil pH, $Ca^{2+}$, $Mg^{2+}$, $Al^{3+}$, $Na^+$, $K^+$, $P^+$, H+Al, OC, OM, air humidity, air temperature, and luminosity.

Demographic data were used to construct an integral projection model (IPM) to calculate population vital rates using continuous variables rather than size classes, as in matrix models (*Easterling, Ellner & Dixon, 2000*). The IPM describes changes during a discrete period in a population whose structure is characterized by a continuous size variable, usually diameter or height. The initial population size is described by a probability density function $n(x, t)$, which represents the proportion of individuals of size $x$ at time $t$. Thus, the model for calculating the proportion of individuals of size y at time $t + 1$ is defined as follows:

$$n(y, t+1) = \int_\Omega \left[ p(x, y) + f(x, y) \right] n(x, t) \, dx$$
$$= \int_\Omega k(y, x) \, ] \, n(x, t) \, dx$$

where $k(y, x)$ represents all the probabilities of transition from an individual size $x$ at time $t$ to an individual size $y$ at time $t + 1$, including new recruits. The *kernel* function is integrated over a set of all size possibilities ($\Omega$) and has two components: the survival and growth function $p(x, y)$ and the fertility function $f(x, y)$. The fertility function is positive for reproductive individuals at time $t$ (parents, $x$) and small individuals at time $t + 1$ (offspring, $y$), obtaining the value 0 for all the other individuals. The function $p(x, y)$ incorporates the growth and survival of all individuals (*Easterling, Ellner & Dixon, 2000*). The survival-growth function is composed of two components:

$$p(x, y) = s(x) g(x, y).$$

**Table 1  Characterization of the sites where populations of *S. coronata* were sampled in Águas Belas, Pernambuco, northeast Brazil.** The populations were submitted to low, intermediate and high frequencies of leaf harvest.

| Harvest Frequency | Site | Coordinates | Distance from Fulni-ô main village | Vegetation | Landscape characteristics | Landscape management history |
|---|---|---|---|---|---|---|
| | Site 1 (L1): Baixinha | 24L0703071/ UTM 8995449 | 5.5 km | Vegetation of arboreal Caatinga and highland swamp. | The area had rocky and clayey soils. | Although not used for animal husbandry or agriculture, it was utilized for harvesting firewood. No fires were reported in the area. |
| Low | Site 2 (L2): Cigana | 24L0719119/ UTM 8992056 | 14 km | Vegetation of Caatinga. | The soil was sandy and had rocky and sandstone compositions. | The area was used for agriculture, itinerant cattle grazing and firewood harvesting. Fires were also not recorded. |
| | Site 3 (I3): Serra Nova | 24L0714343/ UTM 8993046 | 11 km | Vegetation of arboreal Caatinga. | The soil had sandy texture and rocky outcrops. | The area is used for agriculture, including cattle and goat grazing, and firewood harvesting. Fires were recorded. |
| Intermediate | Site 4 (I4): Chiqueiro dos Bodes | 24L0706641/ UTM 8991785 | 18 km | Vegetation of arboreal Caatinga and highland swamps. | The soil had sandy texture and rocky outcrops. | Although the area is not used for growing crops and plantations, it serves as pasture for cattle and is occasionally used for firewood harvesting. No fires were recorded in the area. |
| | Site 5 (H5): Fazenda Nova | 24L0715760/ UTM 8991481 | 11 km | Vegetation of arboreal Caatinga and highland swamps. | The soil had sandy texture and rocky outcrops. | There was a low density of goats. Firewood was harvested at this site. There were fire records. |
| High | Site 6 (H6): Espingarda | 24L0719123/ UTM 8992696 | 15 km | Vegetation of arboreal Caatinga and highland swamps. | The soil was sandy and clayey texture with rocky outcrops. | The area was used for agriculture, including goat grazing and firewood harvesting. Fire was recorded at the site. |

The $s(x)$ component represents the probability of survival of an individual of size $x$, and the $g(x, y)$ component represents the probability of growth from size $x$ to size $y$. The fertility function is also composed of two components:

$$f(x, y) = f1(x)f2(x, y).$$

The component $f1(x)$ represents the average number of offspring produced by an adult of size $x$, which was estimated by the ratio of the number of reproductive individuals per plot in one year to the number of seedlings recruited in the next year (Table S4). In the second component, $f2(x, y)$ represents the probability that an adult of size $x$ will produce an offspring of size $y$.

We calculated alternative statistical relationships for growth and survivorship (vital rates) as functions of plant size (independent, size, size$^2$ size$^3$). Additionally, we subsequently applied model selection methods based on the Akaike information criterion (AIC) to determine which model provided the best fit to the data. We tested the following functions: vital rate~1, vital rate~size, vital rate~size + size$^2$, and vital rate~size+ size$^2$+ size$^3$. To represent the final results of the probabilities for each vital rate calculated for the kernel K, graphs were constructed for each interval and sampled population. Furthermore, we

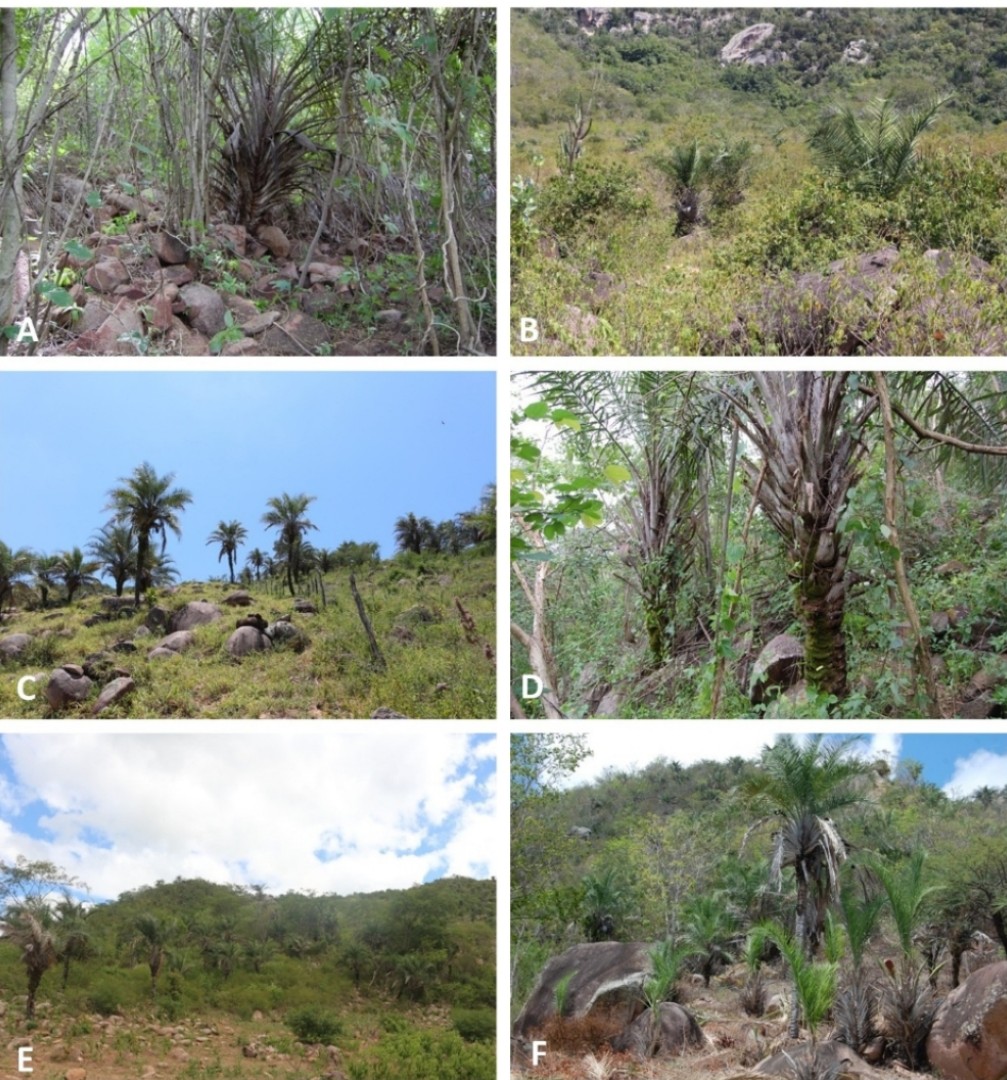

**Figure 2** **Sites where plots were established to collect data on the population structure and dynamics of *S. coronata* in Águas Belas, Pernambuco, northeast Brazil.** (A) and (B) show populations subjected to low-frequency leaf harvest; (C) and (D) show populations subjected to intermediate-frequency leaf harvest; and (E) and (F) show populations subjected to high-frequency leaf harvest. Photo credit: Juliana Loureiro Almeida Campos.

selected the model with the lowest AIC score or the simplest model, using $\Delta$AIC $\leq 2$ (Tables S2 and S3).

As some populations had very few individuals, the two populations with the same leaf harvest frequency were grouped together. In this sense, a model was built by time interval for each frequency of leaf harvest, totaling nine models (three different harvest frequencies and three time intervals). The IPM was constructed based on the total height, as all individuals (seedlings, juveniles and adults) had this measurement. The deterministic population growth rate ($\lambda$) was obtained through the *kernel* for each of the harvest frequency and time

intervals to establish trends in population growth. Thus, $\lambda < 1$ indicates that the population is declining, $\lambda = 1$ indicates stability, and $\lambda > 1$ indicates that the population is growing (*Caswell, 2000*; *Caswell & Fujiwara, 2004*).

Prospective perturbation analysis of elasticity was performed for each sampling interval to verify which individuals in the population and which vital rates had the greatest influence on the population growth rate (*De Kroon et al., 1986*).

$$e\left(y_0, x_0\right) = \frac{K\left(y_0, x_0\right) s\left(y_0, x_0\right)}{\lambda}$$

where $e\left(y, x\right)$ is equal to the integral of 1 and can be interpreted as the proportional contribution of $K\left(y, x\right)$ to population growth (*Ellner & Rees, 2006*). In the equation above, $s(y, x)$ represents the sensitivity formula

$$s\left(y, x\right) = \frac{v\left(y\right) w\left(x\right)}{\left(v, w\right)}$$

where $\langle v, w \rangle$; is the product $\langle v, w \rangle = \int x v(x) w(x) dx$ (*Ellner & Rees, 2006*).

Confidence intervals for population growth rates with 95% confidence intervals were calculated using bootstraps, with 2,000 replicates. In each resampling, 10% of the individuals from each population and time interval were randomly removed.

Life table response experiments (LTRE) were performed to compare the population dynamics under different harvest frequencies (low *vs.* intermediate and low *vs.* high) using the following equation:

$$\lambda t - \lambda c = \Sigma(Kt - Kc)(\partial\lambda/\partial|km)$$

where $\lambda t$ corresponds to the population growth rate under exploitation pressure (intermediate or high) and $\lambda c$ to the population growth rate without exploitation pressure (low). Kt and Kc are the kernels corresponding to the populations under exploitation pressure and without exploitation pressure, respectively. Finally, $(\partial\lambda/\partial K)$ is the sensitivity of $\lambda$ to the perturbation of the mean kernel element (km) (*Merow et al., 2014*).

The statistical analyses were performed in the R 4.2.3 environment (*R Development Core Team, 2023*) using the packages *IPMpack* (*Metcalf et al., 2013*), *fields* (*Nychka et al., 2015*), *gamm4* (*Wood, Scheipl & Wood, 2017*) and *popbio* (*Stubben, Milligan & Nantel, 2008*).

## RESULTS

### Harvesting regimes

Based on interviews with 27 harvesters, the preferred individuals for leaf harvesting were those with the largest and widest leaves. Harvesters typically remove all leaves except for two or three of the youngest leaves. Respondents reported an average interval between harvesting events on the same palm ranging from at least 6 months up to a maximum of 3 years. The mean number of leaves harvested annually per individual was 2.8 in low-frequency harvest sites, 3.61 in intermediate-frequency sites, and 5.14 in high-frequency sites (Table S1).

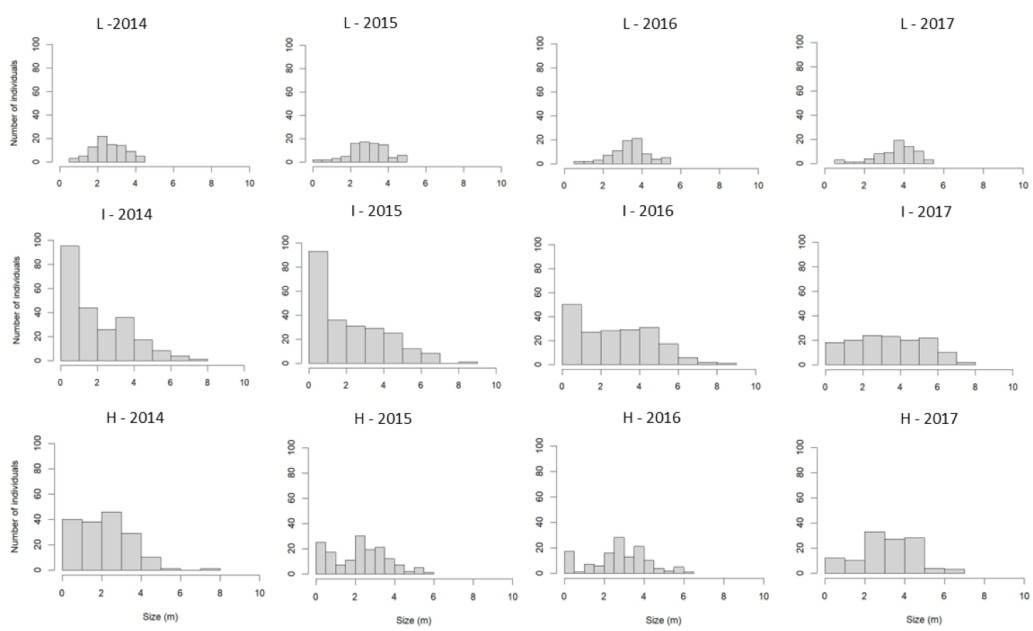

**Figure 3  Size-class distribution of *S. coronata* individuals across four study years (2014–2017) in Águas Belas, Pernambuco, northeast Brazil.** Populations subjected to low-frequency leaf harvest (L), intermediate-frequency leaf harvest (I), and high-frequency leaf harvest (H). $X$-axis = size (m); $Y$-axis = number of individuals.

## Population structure

A total of 549 *S. coronata* individuals were sampled between 2014 and 2017, with an average density of 366 individuals per hectare. Population abundance varied substantially among study sites. The highest abundance occurred in intermediate-harvest-frequency populations ($n = 273$, 546/ha), followed by high-frequency ($n = 190$, 380/ha) and low-frequency ($n = 86$, 172/ha) populations. The tallest individuals were recorded in intermediate-harvest-frequency populations, followed by high and low-frequency populations (Fig. 3). Populations with intermediate harvest frequency showed height class distributions closest to a reverse-J shape in nearly all censuses except the final one (Fig. 3).

## Characterization of leaf harvest sites

The first two principal component axes (eigenvalues = 5.397 and 2.978) collectively explained 86% of the environmental variation in the study area (Fig. 4). Temperature, soil pH, and H+Al showed the strongest loadings on principal component 1 (PC1), while potassium was most influential on principal component 2 (PC2) (Fig. 4). These variables were highlighted as they exhibited eigenvector values ≥0.70. Luminosity and Mg$^+$ were excluded from interpretation due to eigenvector values <0.6 on both PC axes.

The two sites with low-harvest-frequency populations showed substantial environmental and soil differentiation. Soil pH and air temperature were particularly important in sites with the lowest harvesting frequencies, whereas soil nutrients played a greater role in sites experiencing intermediate and high harvest frequencies.

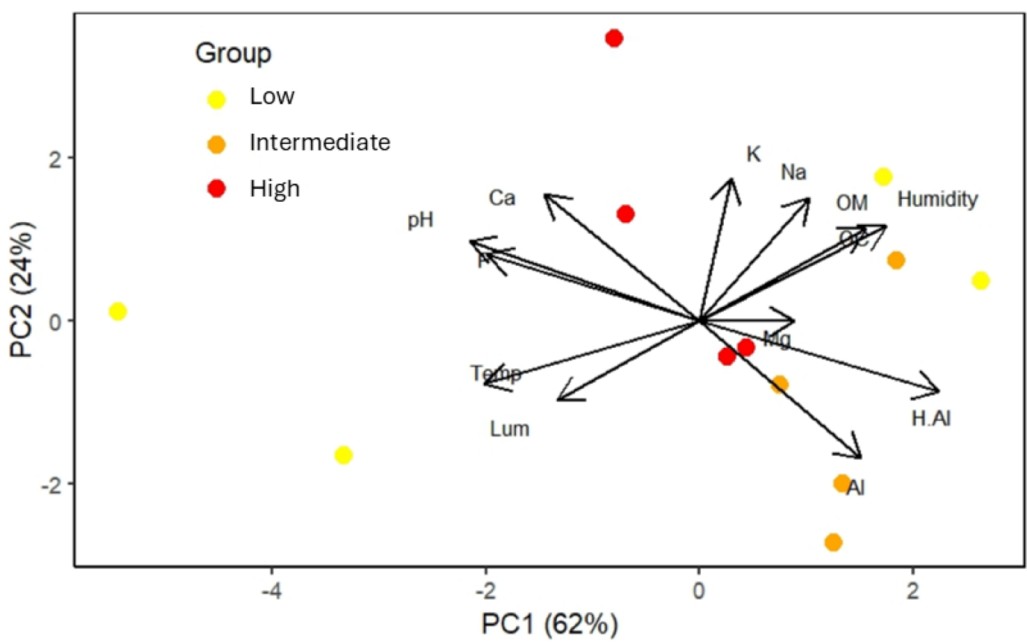

**Figure 4** Order of the twelve plots, located in areas with low, intermediate, and high frequencies of *S. coronata* leaf harvesting by Fulni-ô Indigenous People in Águas Belas, Pernambuco, northeast Brazil. The colored circles represent the plots, with colors indicating harvesting frequency. Abbreviations: Al = Al+; H.Al = H+Al; OM = organic matter; Na = Na+; K = K+; Ca = Ca+; P = P+; pH = pH; Mg = Mg+; OC = organic carbon; Humidity = air humidity; Temp = air temperature; Lum = Luminosity.

### Influence of leaf harvest on the population dynamics of *Syagrus coronata*

All populations decreased in the last year of harvest but exhibited different trends between the first and third time intervals. The populations subjected to low leaf-harvesting frequency grew at the second sampling, decreasing sharply at the third one (Table 2). Populations with intermediate frequency of leaf harvest decreased in all sampling intervals, while highly harvested populations increased in the second time interval, then decreased in the next one (Table 2). Moreover, they had the lowest population growth rates ($\lambda$).

The IPM analyses revealed that individuals from populations subjected to low leaf harvest frequency exhibited equal growth probabilities during the first and second time intervals. In the third time interval, however, intermediate and large-sized individuals showed greater growth compared to small individuals (Fig. 5). Small individuals from populations subjected to intermediate leaf-harvesting frequency showed greater height growth in the first and second time intervals, but experienced higher mortality rates in the second and third intervals. In these populations, adult size did not change significantly. Among the populations subjected to the highest harvest frequencies, the younger individuals experienced greater height growth during the first time interval, but their mortality rate was also high. Taller individuals had a lower height growth rate and a lower mortality rate at all time intervals (Fig. 5).

**Table 2  Population growth rate (λ) with 95% confidence intervals (CI−, CI+) for three populations of *S. coronata* under varying leaf-harvest frequencies in Águas Belas, Pernambuco, northeast Brazil.**

| Population | Sampling interval (years) | λ | CI− | CI+ |
|---|---|---|---|---|
| | 1 | 1.87 | 1.8423 | 1.9010 |
| Low | 2 | 1.90 | 1.8952 | 1.9148 |
| | 3 | 1.12 | 1.1221 | 1.1229 |
| | 1 | 1.02 | 1.0183 | 1.0200 |
| Intermediate | 2 | 1.00 | 0.9997 | 1.0006 |
| | 3 | 0.89 | 0.8843 | 0.8956 |
| | 1 | 0.94 | 0.9416 | 0.9459 |
| High | 2 | 0.99 | 0.9924 | 0.9943 |
| | 3 | 0.96 | 0.9636 | 0.9661 |

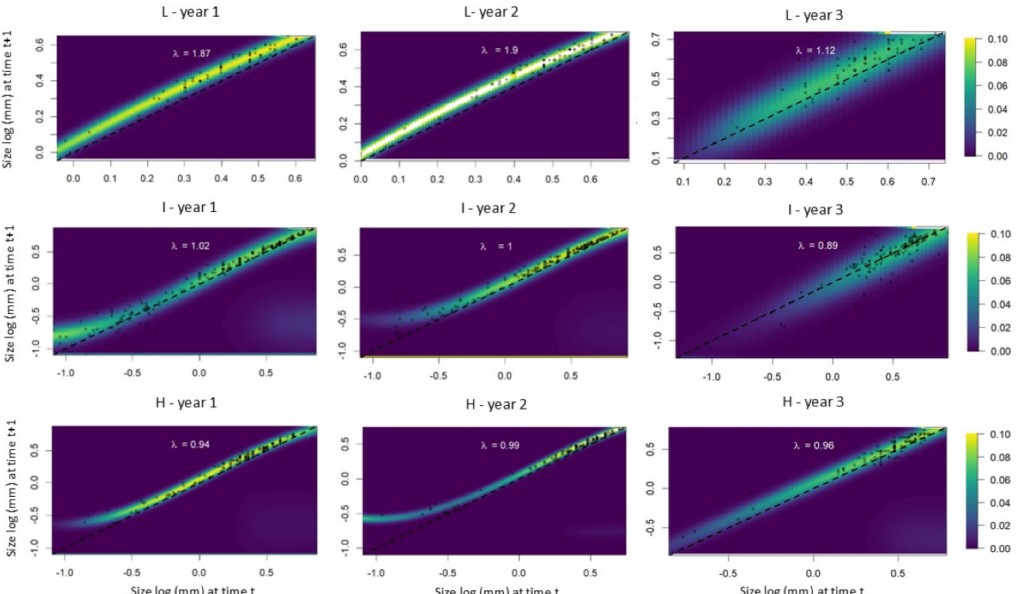

**Figure 5  Probability of growth, stasis, regression P(x,y) and fecundity F(x,y) as a function of height for *S. coronata* individuals subjected to low (L), intermediate (I) and high (H) leaf harvest frequencies.** The lowest and highest probabilities are shown in blue and yellow, respectively. The *x*- and *y*-axes are on a logarithmic scale.

The elasticity analyses showed that, in populations subjected to lower harvest frequencies, the growth of smaller individuals had a greater influence on the population growth rate (Fig. 6). Conversely, in populations subjected to intermediate and high harvest frequencies, the presence of larger individuals had a greater influence on the population growth rate (Fig. 6).

The LTRE analyses showed that the stasis of smaller individuals contributed positively to the increase in the population growth rate in the populations subjected to intermediate

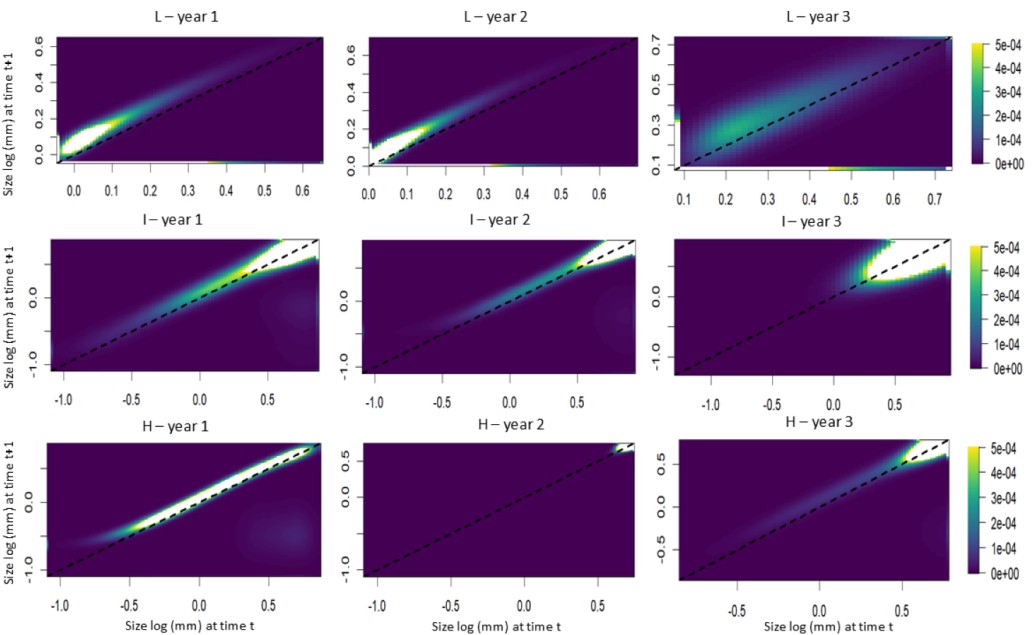

**Figure 6** **Elasticity analysis as a function of transition probability (height) for *S. coronata* populations subjected to low (L), intermediate (I) and high (H) frequencies of leaf harvest in Águas Belas, Pernambuco, northeast Brazil.** The lowest and highest elasticities are shown in blue and yellow, respectively. The *x*- and *y*-axes are on a logarithmic scale.

harvest frequencies. However, in these same areas, the increase in height of individuals of intermediate size contributed negatively to the population growth rate. Among the populations subjected to the highest harvest frequencies, the presence of larger individuals contributed positively to the first time interval, while the growth of small and intermediate individuals contributed negatively. In the second interval, the stasis of smaller individuals and the growth of intermediate individuals negatively contributed to the population growth rate. In the third interval, the growth and regression of the intermediate individuals negatively contributed to the population growth rate (Fig. 7).

## DISCUSSION

Results showed that *S. coronata* populations subjected to high leaf harvest frequencies decreased in size over the sample years. Leaf harvest often leads to population decline if it involves the felling and death of individuals (*Portela, Bruna & Maës dos Santos, 2010*) or if it is performed on young individuals (*Svenning & Macia, 2002*; *Mendes, Galdino & Portela, 2022*). However, demographic studies of palm trees whose leaves are harvested have already shown decreased population growth rates compared to those in control areas (*Endress, Gorchov & Berry, 2006*; *Valverde, Hernandez-Apolinar & Mendoza-Amaro, 2006*). The elasticity analysis revealed that the survival of the largest individuals was vital for the growth of the populations subjected to intermediate and high harvest frequencies. Therefore, these individuals must be maintained to ensure the stability of these populations.

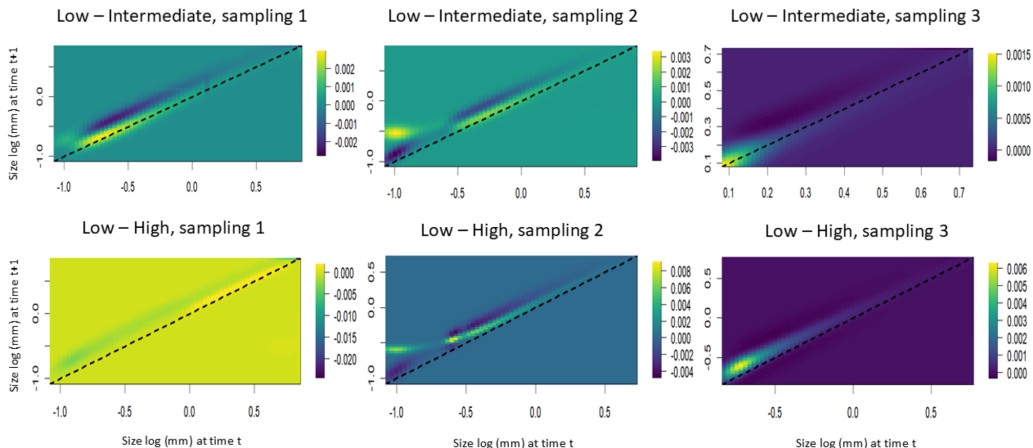

**Figure 7** **Life table response experiments (LTRE) for *S. coronata* populations subjected to low, intermediate and high leaf harvest frequencies in Águas Belas, Pernambuco, northeast Brazil.** The lowest and highest elasticities are shown in blue and yellow, respectively. The *x*- and *y*-axes are on a logarithmic scale.

In long-lived species that exhibit multiple reproductive events over several years, adult survival plays a key role in population stability (*Silvertown, Franco & Menges, 1996*; *Horvitz, Schemske & Caswell, 1997*), as shown in our study. Adults are important because they contribute to population fertility. However, as adults become older, population persistence will depend not only on young individuals transitioning to the reproductive stage but also on continuous recruitment.

Other characteristics that may be related to the low survival of adults in populations subjected to intermediate and high harvest frequencies include soil nutrients, since some soil variables were particularly important in the areas where these populations occur, as shown here. However, no studies have investigated the relationship between edaphic characteristics and adult survival in *S. coronata*. It is clear that adult survival in these sites is low, and leaf harvest may contribute to this pattern, as interviews revealed that the Fulni-ô preferentially harvest large juveniles and adults due to their leaf size. Previous research has shown that among the Fulni-ô, more experienced artisans harvest leaves more sustainably than less experienced ones. Specifically, experienced artisans avoid removing the newest leaves from palm trees (*Campos et al., 2018*). Therefore, we recommend sharing this traditional knowledge throughout the community.

Recruitment, measured by the establishment of new individuals between sampling intervals, was lower in populations with the lowest harvest frequencies compared to those with intermediate and high frequencies. Surprisingly, despite lower recruitment, these populations showed the highest growth rates. A similar pattern was reported by *García, Galeano & Bernal (2017)* for *Astrocaryum malybo* palms harvested for handicrafts in Colombia, where populations grew despite low seedling proportions. This pattern likely results from the limited influence of recruitment and fecundity on population growth rates, as revealed by elasticity analysis. Rather, survival and growth of smaller individuals

(≤2.5 m) contributed substantially to population growth. The high λ values observed in these populations (Fig. 6) reflect successful growth and survival of these smaller individuals between sampling periods. Maintaining this dynamic is essential for population persistence.

Despite these findings, the decline in λ values across sampling years is notable, and the low fertility in populations with low harvest frequencies should serve as a warning. The lack of recruitment suggests these populations may eventually decline as their structure becomes dominated by adult individuals (*Hall & Bawa, 1993*; *Lykke, 1998*; *Sá, Scariot & Ferreira, 2020*). Previous studies demonstrate that leaf harvest reduces reproductive output through photosynthetic compensation (*Mandle, Ticktin & Zuidema, 2015*; *Lopez-Toledo et al., 2018*), where photoassimilates are allocated to new leaf production rather than to inflorescences and infructescences (*Anten, Martínez-Ramos & Ackerly, 2003*). Therefore, while greater recruitment might be expected in lightly harvested *S. coronata* populations due to increased reproductive structure production, this was not observed.

The environmental conditions at low-harvest-frequency sites may have contributed to both low adult fecundity and limited recruitment of new individuals. Our PCA results highlighted air temperature as an important factor, consistent with studies demonstrating negative impacts of extreme climate on tropical plant demography (*Wright, 2005*; *Martínez-Ramos, Anten & Ackerly, 2009*). Additionally, key soil characteristics - including texture, drainage, nutrient availability, and pH - may have restricted seed germination and seedling survival, as observed in other palm species (*Eiserhardt et al., 2011*).

Although not directly measured in our study, cattle and other livestock in the study areas may have further reduced recruitment through trampling and herbivory of smaller plants (*Fleury et al., 2015*; *Hordijk et al., 2019*). This pattern has been documented in other tropical palm species subject to leaf harvest (*Brokamp et al., 2014*; *Torres, Galeano & Bernal, 2016*). In the Caatinga specifically, agricultural and livestock activities have decreased *S. coronata* recruitment, with livestock trampling being a primary contributing factor (*Pereira et al., 2021*; *Lima, Scariot & Sevilha, 2023*). These palms primarily occur in "serras" - high-altitude marshes that originally supported continuous vegetation cover. Many of these areas are now occupied by non-indigenous families through land leasing arrangements initiated by some Fulni-ô community members to generate income. Those Fulni-ô who oppose leasing view it as a major threat to *S. coronata* populations (*Campos et al., 2018*), as the leased lands are used for agriculture and livestock (particularly cattle and goats), substantially altering original land use patterns.

The observed population declines in the final sampling interval (despite some populations maintaining λ ≥ 1) highlight the need for management strategies that promote both stabilization and growth while accounting for interannual fluctuations. Addressing this will require long-term demographic monitoring, as palm species typically show cumulative responses to leaf harvest (*Guilherme et al., 2015*), despite some documented short-term effects (*Calvo-Irabién, Zapata & Iriarte-Vivar, 2009*).

## LIMITATIONS OF THE STUDY

We acknowledge that the lack of a control area (with no harvesting) limited our conclusions regarding the factors that drive the demographic response of *S. coronata*

populations. Without a control area, it is difficult to determine whether harvesting alone or environmental factors are responsible for the observed population patterns. Despite this limitation, studying multiple sites for each of the three harvesting frequencies helped us better understand the influence of this factor on the species' demography. Another factor that somewhat limited our conclusions was the small size of the studied populations, though we believe this was mitigated by conducting data collection over four years (across three time intervals), which provided more IPMs for interpretation.

We also recognize that the lack of information about past land management conditions, such as grazing and burning practices, makes it difficult to draw definitive conclusions about the factors that best explain the current demographic characteristics of the populations. We recommend that future studies incorporate historical land-use data to clarify how these practices may have shaped population structure over time.

While we cannot state with complete certainty which factors are primarily responsible for the demographic responses of *S. coronata*, our study makes important contributions by demonstrating that maintaining individuals of different sizes is crucial for population viability. Furthermore, this study helped identify the most suitable areas for leaf collection by the Fulni-ô Indigenous People, supporting both their extractive activities and handicraft production—practices that represent traditional knowledge and an essential way of life. By including environmental variables in our study, we have shown that these factors are important drivers and should be considered in studies examining harvesting effects on plant demography.

To address these limitations, we recommend that future studies in this field include control areas in their datasets. Another suggestion is to implement long-term population monitoring.

## CONCLUSION

Our study showed the importance of individuals of different sizes for maintaining *S. coronata* populations, whose leaves are used by the Fulni-ô Indigenous People in northeast Brazil. In populations with lower harvest frequencies, the growth of smaller individuals most influenced population growth rates, while in populations with intermediate and high harvest frequencies, larger individuals were most important for population growth. Despite exhibiting the highest population growth rates, populations subjected to low harvesting frequencies showed limited seedling recruitment and concerning demographic structure. Based on our results, we recommend the following management practices for *S. coronata* populations in collaboration with the Fulni-ô Indigenous People:

(i) Maintain current leaf harvest in areas with low harvest frequencies, as these populations showed high growth rates.

(ii) Protect smaller individuals (under 2.5 m) in low harvest frequency areas. Although their growth contributed significantly to population growth rates, the low recruitment observed highlights the need to protect seedlings and understand what limits new individual establishment in these areas.

(iii) Preserve intermediate-sized (2.6–5 m) and larger individuals (over 5 m) in areas with intermediate and high harvest frequencies. The survival of these size classes, which

produce better quality leaves for Fulni-ô handicrafts, was most important for population growth in these declining populations. We suggest alternating harvest years in these areas (*e.g.*, harvesting every other year).

(iv) Develop agreements between the Fulni-ô and tenant farmers on Indigenous lands. Farming practices can make environments less suitable for palm establishment. Vegetation clearing and livestock trampling may have contributed to low recruitment in Site 2 (Cigana), where harvest frequency is lower.

## ACKNOWLEDGEMENTS

The authors would like to thank the Fulni-ô Indigenous People of Águas Belas for their contribution to the development of this research and the members of the Laboratory of Ecology and Evolution of Socioecological Systems (LEA) at Federal University of Pernambuco (UFPE) for the support provided during data collection. We acknowledge Camila dos Santos de Barros for the support provided in the model's interpretation.

### Funding

This work was supported by the National Council for Scientific and Technological Development (CNPQ), and "Projeto Bem Diverso", executed by Embrapa Genetic Resources and Biotechnology, United Nations Development Programme (PNUD) with resources of the Global Environment Facility (GEF). The funders had no role in study design, data collection and analysis, decision to publish, or preparation of the manuscript.

### Grant Disclosures

The following grant information was disclosed by the authors:
National Council for Scientific and Technological Development (CNPQ).
Embrapa Genetic Resources and Biotechnology, United Nations Development Programme (PNUD).
Global Environment Facility (GEF).

### Competing Interests

Ulysses Paulino Albuquereque is an Academic Editor for PeerJ.

### Author Contributions

- Juliana Loureiro Almeida Campos conceived and designed the experiments, performed the experiments, analyzed the data, prepared figures and/or tables, authored or reviewed drafts of the article, and approved the final draft.
- Elcida de Lima Araújo conceived and designed the experiments, authored or reviewed drafts of the article, and approved the final draft.
- Aldicir Scariot conceived and designed the experiments, authored or reviewed drafts of the article, and approved the final draft.
- Eduardo Teles Barbosa Mendes analyzed the data, authored or reviewed drafts of the article, and approved the final draft.

- Rita de Cássia Quitete Portela analyzed the data, authored or reviewed drafts of the article, and approved the final draft.
- Ulysses Paulino Albuquerque conceived and designed the experiments, authored or reviewed drafts of the article, and approved the final draft.

## Human Ethics

The following information was supplied relating to ethical approvals (i.e., approving body and any reference numbers):

The project was approved by the National Commission of Ethics in Research (CAAE 24211014.0.0000.5207), National Indian Foundation (authorization No. 04/AAEP/PRES/2015), National Historical and Artistic Heritage Institute (case No. 2000.000203/2014–35), which are responsible for approving research with traditional communities and Indigenous Peoples in Brazil.

## Data Availability

The data is available in the Supplemental Files.

## Supplemental Information

Supplemental information for this article can be found online at http://dx.doi.org/10.7717/peerj.19739#supplemental-information.

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
