# Peer review of "The importance of individuals of different sizes in the population maintenance of a palm species used by the Fulni-ô Indigenous People in northeast Brazil"

_PeerJ, doi:10.7717/peerj.19739_

## Round 0.1 · original submission · Major Revisions

The importance of individuals of different sizes in the population maintenance of a palm species used by the Fulni-ô indigenous people in northeastern Brazil.
2024:03:98566
Please accept my sincere apologies for the exceptional delay in reviewing your submission. It was challenging to find available reviewers who did not have conflicts of interest with any of the authors.
I have now received the comments from three expert reviewers all of which agree the study is interesting. The reviewer comments are clear so I will not repeat them here. I agree with reviewers 1 and 3 that both methods and results need major revisions. My main concern is that with the low number of sample sites and the current analysis/results it is not clear whether harvest is a driver or whether other environmental factors are also influencing patterns in the populations. Size appears to be important but is this due to harvest or environment or an interaction? Authors may consider exploring use of individual based models that explicitly consider harvests and environmental factors and the associations over time. Indeed, it is not clear whether 3 years provides sufficient time to robustly evaluate populations of this species? Depending on the revisions made the authors may need to revise the text including the conclusions to more appropriately reflect the data collected (lack of any sort of experimental design – i.e. there are no controls such as areas without harvest) and the results presented.

·

Basic reporting

.

Experimental design

I think the authors need to provide more information on the harvesting regimes at the individual level.
The populations used for the study have been always harvested in the very same intensity or were these treatments created/followed up for this study?
I believe people harvest depending on the market and on the availability of the leaves.
Do the regimens imply that individuals from low-intensity populations are harvested 12 times a year, mid are harvested 24 times and high intensity are about ~52 times?
How many leaves do the people harvest at each event?
Is there any estimation of how many leaves people harvest at a single plot from the different harvesting regimes?
At a single plot, all individuals are harvested each visit? I mean if a single individual in a low-intensity plot is harvested 12 times, at middle intensity a single individual is harvested 24 times and so on?

The harvesting regimes may not exist and maybe the populations respond as a function of the environmental/anthropic factors.

Validity of the findings

Very important and very valuable findings.

Additional comments

I understand that sometimes is difficult to find the right locations/conditions for field experiments. Also, I understand that populations are subjected to a multiple set of environmental/anthropic conditions. If we do not set an experiment up separating the major driver of population dynamics, sometimes is just difficult to make assumptions that only one major factor is driving population dynamics. I believe this case is one of these examples. I´m bit concern on the differences in environmental conditions of the plots used in the study. Some plots had animal husbandry, agriculture or fires, while others not. How confident are we that demographic processes observed are as a response to harvesting. I see authors recognize that sites come from a different environmental and anthropic context. However, I have a slight concern on the main goal of the study. Background and hypothesis mainly focus on how palm population dynamics will respond to leaf harvesting, which I believe is difficult to prove with the present design. Instead, I would provide more information on how the environmental factors/harvesting factors interact and define the dynamics of populations and would twist the goals of the study.
Also, despite the importance of different size classes, I believe there are other important elements to consider in the title such as the environmental/anthropic context.
I would invite authors to consider these opinions and reorganize their manuscript.


Abstract
Line2. “demography” or “demographic processes”, instead of “demographic behavior”?
Line3. The demographic responses of the palm Syagrus ….
Were the population studied harvested? Did the different plots have different harvest regimes?
L190. Can the authors specify the “snowball technique”?
L197. Can the authors give more details of the harvesting regimes? Do these regimens imply that low intensity are harvested 12 times in the year, mid are harvested 24 times and high intensity are about ~52 times.
How many leaves do the people harvest at each event? At a single plot, all individuals are harvested each visit? I mean if a single individual in a low intensity plot is harvested 12 times, at middle intensity a single individual is harvested 24 times and so on?
L204. Did the authors register leaf harvesting for each palm? How confident is the low-mid-high intensity? It would be great to have an estimate of how many leaves are harvested per individual. In some cases, these treatments yield similar number of leaf harvested per individual monthly/annually, because plants are not harvested each single time. For example, some palms at the high-level treatment are not likely harvested four times a month, but once a month which maybe like the low intensity treatment. Palms are not able to recover growth and leaf production, they take some time to functionally recover, and have leaves ready for the next harvesting. Of course, that depends on the species, but some may take even a year or more to recover. That means, they do not have leaves at each harvesting event and they´re not harvested (see Lopez-Toledo, L. et al. 2012. Journal of Ecology 100: 1245-1256).
Is there any way authors can include some information or inquire with local people?

Figures
Figure 5. What do the authors mean with “(including white)”
Figure 6.
The lowest and highest

Reviewer 2 ·

Basic reporting

No comment.

Experimental design

No comment.

Validity of the findings

No comment.

Additional comments

The research findings, conclusions, and recommendations are relevant to enhancing Arecaceae leaf harvest management and promoting sustainability. The graphs are very informative and adequate to the data. The methods are described clearly and in detail. The discussion section could be strengthened by including additional Arecaceae species used for leaf harvesting. I have recommended some papers in the comments on the manuscript.

Annotated reviews are not available for download in order to protect the identity of reviewers who chose to remain anonymous.

Reviewer 3 ·

Basic reporting

The document is well-written with minor issues that are reported below. The introduction provided sufficient background information.

The document was structured following the rules of the journal and was clear.

The results presented were relevant to answer the research question.

Experimental design

The research is within the aims and scope of the journal
The research question was well defined and stated a knowledge gap.
The authors presented ethical and technical support information on how the work in the field was going to be handled.
The methods section needs a lot of improvement. They do not explain how the state variable of height was measured in each individual and how the size increases were measured. In the ethods section

It would be beneficial to give the context on the demography of how much an individual is harvested in one year. ie: 3 leaves per year. Since the harvest intensity is defined as visits to a place on a monthly/weekly basis it is not clear from the text if they harvest all of the individuals or just some of the individuals in the area.

Another aspect that is not covered and would improve the analysis is the individual’s harvest size. It is most likely that the Fulni-ô harvest individuals within a certain size range to cover the fiber requirements.

It is important to explain how the authors measured the vital rates of growth of individuals. What state parameters do they use and how do they estimate the yearly increase? It wasn´t explained in the main document and is essential for reproducing the IPM models presented.

In many parts of the document, the authors refer to adults as an important category without actually defining the category. ¿At which height a Syagrus coronata individual starts reproducing?

Validity of the findings

The information regarding the IPM in the document and the supplementary material does not allow to assess the quality of the models presented in this work. However, the work has great potential to be a meaningful contribution to the effects of leaf harvest in plant vital rates, if the missing information is provided.

The IPM models rely on generalized linear models best fitted to the data to show meaningful results. The authors only provided the AIC which by itself does not allow us to understand if a model selected has a good fit. Other statistics (R2, P, Residual standard error, maximum likelihood estimation ) have to be presented to assess the predictive power of the models selected, and therefore the IPM quality to make inferences about the population dynamics.

The explanations provided for the results are well-stated and linked to the original research question arising from the presented results.

Additional comments

Lines 79-80. The authors write, “However, no impacts were reported on survival rates (Zuidema et al., 2007; Hern·ndez-Barrios et al., 2012).” Please note that some studies have found potential or measured impacts on survival rates when populations are subject to leaf harvest.

Lines 155 and 160. If the authors are starting a sentence either at the beginning of a paragraph or after a period the genus of the species has to be written completely; e.g. “Syagrus coronata” should not be abbreviated, even if it was named before.

Lines 178-179. Explain how the height ranges were established. Stem base to highest leaf or to leaf crowns.

Lines 236 and other parts. The authors mention “The vegetation of the arboreal Caatinga plants was in a regular state of conservation.” Please clarify. Is it intended to suggest that the area was in a ‘good’ ecological condition? The term ‘regular’ can be used to describe a normal or frequent state of being, but is not customarily used to describe an area’s conservation status. In addition, “the vegetation of the arboreal Caatinga plants” is a redundancy, vegetation and plants are synonyms

Line: 415. The authors wrote “Recruitment, indicated by the entry of new individuals between the time intervals sampled, was lower in populations subjected to the lowest frequencies of leaf harvest.” Please indicate lower than which other populations intermediate and/or high leaf harvest frequencies.

For Figure 3 the axis labels were not provided in the figure itself but, in the legend. I would suggest including the axis label in the actual figure as presented in all of the other figures representing data.

---

## Round 0.2 · Major Revisions

I have now received comments from two of the original reviewers. I agree that the methods and results still need additional clarification, and that this could affect both the interpretation and insights generated. Additionally, my main concern does not appear to have been addressed - i.e. that with the low number of sample sites and the current analysis/results it is not clear whether harvest is a driver or whether other environmental factors are also influencing patterns in the populations. Whether 3 years is sufficient, and that conclusions should more appropriately reflect the data collected (lack of any sort of experimental design – i.e. there are no controls such as areas without harvest). I suggest authors consider adding a caveats section to the discussion to specifically address these issues for readers. I look forward to receiving your revised submission.

Christie, A. P., Amano, T., Martin, P. A., Shackelford, G. E., Simmons, B. I., & Sutherland, W. J. (2019). Simple study designs in ecology produce inaccurate estimates of biodiversity responses. Journal of Applied Ecology, 56(12), 2742-2754.

Reviewer 2 ·

Basic reporting

No comment

Experimental design

Comment in line 247: Can the difference in leaf count be attributed solely to harvesting by the Fulni-ô community, excluding natural losses like drying, disease, or climatic events such as wind?

Validity of the findings

It is very important for the rational use of this palm tree, the preservation of harvested populations, and the permanence of the Fulni-ô people.

Additional comments

A few corrections have been noted in the PDF.

Annotated reviews are not available for download in order to protect the identity of reviewers who chose to remain anonymous.

Reviewer 3 ·

Basic reporting

No comment

Experimental design

No comment

Validity of the findings

The authors' response raised more questions about the validity of the IPMs. I will point out each one of them:
1. In the text, they stated, "We tested the following functions: vital rate ~1, vital rate ~ size, vital rate ~ size + size², and vital rate ~size + size²+ size³." However, in the supplementary material, they only showed the models of the function of survival in relation to size. Missing, still, are models of the initial size in relation to the final size in order to calculate the probability of individuals growing in the probability density function. Also, the fecundity models of f2 (x,y) were not presented in the text nor in the supplementary material.
2. Equations should be written in mathematical notation, not in r code language, as in any other science document for reproducibility.
3. The parameters (Y=ax +b, a, b) used in each regression model selected to calculate the probability density (n(y,t+1)) should be presented so every reader can understand the behavior of each vital rate of the different populations, which also adds transparency to how the data was managed. The parameters of fecundity f1 calculated from "the ratio of the number of reproductive individuals per plot in one year to the number of seedlings recruited in the next year" should also be included.
4. Some of the linear regression models with the best fit were not good predictors of the data. With low R2 (less than 20% of data explanation), P above 0.05 and a residual standard error that did not change across regressions (see low sampling interval 3, medium sampling interval 2). Some of the models selected were not consistently good predictors of survival in terms of the size of the individual.
4. I would recommend reviewing the following articles to improve the IPMs and, in particular the comments made in the previous points:
Rees, M., Childs, D. Z., & Ellner, S. P. (2014). Building integral projection models: a user's guide. Journal of Animal Ecology, 83(3), 528-545.
Zuidema, P. A., Jongejans, E., Chien, P. D., During, H. J., & Schieving, F. (2010). Integral projection models for trees: a new parameterization method and a validation of model output. Journal of Ecology, 98(2), 345-355.
Martínez-Ballesté, A., & Martorell, C. (2015). Effects of harvest on the sustainability and leaf productivity of populations of two palm species in Maya homegardens. PLoS One, 10(3), e0120666.

Additional comments

No comment

---

## Round 0.3 · Major Revisions

We have received extensive comments from 2 reviewers on the current version of your manuscript. They are both happy to you have made good improvements to your manuscript, but there are still several points that they made to help further imrpove your work. Please try to address reviewer comments as you can and I look forward to seeing a revised version of your manuscript in the future.

**Language Note:** The review process has identified that the English language must be improved. PeerJ can provide language editing services - please contact us at [email protected] for pricing (be sure to provide your manuscript number and title). Alternatively, you should make your own arrangements to improve the language quality and provide details in your response letter. – PeerJ Staff

·

Basic reporting

No comment

Experimental design

I read a previous (maybe the first) version of this manuscript and I believe it has substantially improved and it is almost there. I still notice some missing information on the experimental design, which I suggested in my previous review. I regret they have not a control (with no harvesting) management plot, but at least they have acknowledge in the discussion. However, I believe they also must explain it the in the methods section.
I have some minor comments, which I believe can improve the clarity and scope of their manuscript to make a stronger contribution.

Validity of the findings

No comments

Additional comments

Minor comments
L66. ....as nutritional, but also generating an important income for some communities. Include some other references.
L109. Mauritia flexuosa is source of very important income in the Amazon. Review the works from Virapongse and other collaborators.
L114. Do the authors have bit more information on how local people manage the natural populations, in terms of harvesting intensity and frequency?
L163. Syagrus coronota is called ouricuri by the Fulni-o, people? Can you specify it here?
L194-195. I believe this information should go somewhere in the 151-161 paragraph.
L217-221. In a previous review, I asked the authors to include more info on management harvesting. Can a single individual be harvested four times a month? How many leaves approximately are harvested per individual?
The authors should explain if the experimental plots they used have been used before and for how long. Or the plots were unexploited and they applied harvesting for first time to this populations
I remember authors do not have control plots, right? Can the authors specify it here. Is that because that does not exist? are there no localities without harvesting?
L223. Can the authors specify that the six sites with different leaf harvesting intensity are the same where they monitored the population dynamics?
Were the individuals monitored to register how many leaves were harvested for each one?

Results
I like the nice results and figures.

Do the authors have information on number or % of leaves harvested per single individual? It would be nice to see in this section, at least a brief description on how many leaves are harvested per individual and per plot at each type management.

L473. I recommend to review the work from Lopez-Toledo et al 2012. Journal of Ecology, which describes the effect of chronic harvesting and subsequent recovery after leaf area harvesting.
L531. …. I recommend delete ….. which is not always possible………” We all know those limitations.
L553. I believe you cannot recommend this, yet, but you can suggest it to investigating.

Reviewer 4 ·

Basic reporting

The data presented by the authors of this manuscript are relevant and of interest to a broad audience working in the field of sustainability of non timber forest products harvesting.

As I am not a native English speaker, I apologize in advance if I am wrong. However, I have the feeling that several parts of the text need additional checking of the style, semantics and language used.I think the text could be greatly improved with some additional proofreading.

Many studies on wild palm demography use IPMs, transient dynamics and density-dependent methods. It strikes me that you do not cite these papers. Some of these studies are listed below:

García, Néstor, Gloria Galeano, and Rodrigo Bernal. "Demography of Astrocaryum malybo H. Karst.(Arecaceae) in Colombia, recommendations for its management and conservation." Colombia forestal 20.2 (2017): 107-117.

Nazareno, Alison G., and Maurício S. dos Reis. "At risk of population decline? An ecological and genetic approach to the threatened palm species Butia eriospatha (Arecaceae) of Southern Brazil." Journal of Heredity 105.1 (2014): 120-129.

Isaza, C., Bernal, R., Galeano, G., & Martorell, C. (2017). Demography of Euterpe precatoria and Mauritia flexuosa in the Amazon: application of integral projection models for their harvest. Biotropica, 49(5), 653-664.

Isaza, Carolina, Carlos Martorell, Daniela Cevallos, Gloria Galeano, Renato Valencia, and Henrik Balslev. "Demography of Oenocarpus bataua and implications for sustainable harvest of its fruit in western Amazon." Population ecology 58 (2016): 463-476.

Other relevant article:

Zambrano, J., & Salguero‐Gómez, R. (2014). Forest fragmentation alters the population dynamics of a late‐successional tropical tree. Biotropica, 46(5), 556-564.


This work is relevant for achieving a better understanding of the sustainability of harvesting in a family as important to people as the Arecaceae family.

I believe the data are valuable. The study integrates ecological and social approaches to improve the knowledge about S. coronata. However, I think there are still many changes to be made. Although the previous reviewers did not request them, the changes are necessary to obtain an accurate picture of the information that will be published.

Experimental design

Although it would have been desirable to have more plots per treatment or condition, the size of the plots is evidence of a laborious work. Authors used standardize and reproducible methods.

Perhaps there are details of construction of the kernels that are not enough clear.

Two or three years of monitoring will always be considered too short due to the long-lived characteristics of palms. Nevertheless, it is important to obtain this overview of different management systems and their impact on the conservation of a highly valuable biocultural resource such as S. coronata. Therefore, this study is relevant.

The the lack of a unharvested systems should not be considered a weakness of the study system. That is the reality of useful species in rural contexts, where they are a resource for people. However, one serious problem may be that you start with very heterogeneous and (which is even worse) unknown management conditions in the past. Before installing the study, how many times could sites 1 and 2 have been burned or grazed to lack individuals over 6 m long and seedlings nowadays? From the beginning, this suggests that it is not precisely (or not only) the harvest that explains the general ecological traits of these populations.

Validity of the findings

It is noteworthy that environmental variables were included in the analysis. However, it is striking that air temperature is highlighted as an explanatory factor for site variability when the sampling units are located in an area smaller than 30 km². Unless there are landscape features leading to significant differences in winds or slopes. If so, this should be explained.
Perhaps the time differences in which the data were taken or the differences in the seasons (rainy/dry) could better explain this variation and not the sites as such. The same is true for factors such as potassium or magnesium. This is not to say that these differences do not exist.
However, if the population structures are examined closely, it is possible that other factors (historical/contemporary) are responsible for the differences between the sites.
It is odd (so to speak) that the sites with lower harvests frequencies, where there is apparently less anthropogenic intervention, have population structures that are the least similar to what would be expected for a natural population.

Although the parameterization of the vital rates is found in the supplementary materials, it is necessary to show how the best-fitted models match the reality of the data collected in the field. To do this, what is usually done in demographic studies is to plot the behavior of the data observed in the field against the selected model. This would provide transparency on the robustness of the selected models. It can also show a proxy of the fecundity probabilities for example with the probability of individuals to flower or fruit.

The value of the LTRE and elasticity analyses is that they make a diagnosis of the contribution of each demographic process to the differences in population-growth rate between different treatments. Although the authors present a graph of the “transition”, it is not clear why they do not present figures for another parameters such as fertility. It would be absolutely possible to do.
It is also unclear whether there are differential contributions between individuals of different sizes. It is also uncertain why authors did not provide results for the comparison of low vs. high harvest rates.

A projection showing how populations would respond to the mortality of a certain percentage of reproductive individuals (e.g., 5% to 20%) may reveal how much growth rates would decline if there were no new recruits.

Regarding conclusions.

The study data do not support the conclusion that populations with individuals of different sizes better guarantee the viability of such populations. Populations showing the best lamda have the worst population structure.

Perhaps the population growth rates obtained are not a result of the harvest frequency. The number of kernel divisions could have influenced the weight given to growth in the IPM. Thus, populations with shorter-stemmed individuals (Fig. 3) performed better because more of their individuals (proportionally speaking) transitioned to another category during the years of observation.

Additional comments

I have additionally commented in several parts of the pdf.

Please, consider mentioning at least the main use of S. coronata leaves in the abstract.
In the section "Additional information about Syagrus coronata leaf harvesting regimes", please, explain which palm, no leaves, sizes are harvested.

In order to observe differences transparently, the y-axis of plots 5 and 6 must be the same in all cases. Unless the selection of size ranges had been done differently when you constructed the subkernels for the low, medium and high harvest rate populations. If so, this should be explained in the methods section.

It would also be highly desirable to have a figure of mortality rates, in supplemental files.

The supplemental material should include figures and tables with headings. The appropriate mathematical notation should also be used in the columns σ(x) and γ(x, y).

Annotated reviews are not available for download in order to protect the identity of reviewers who chose to remain anonymous.

---

## Round 0.4 · accepted · Accept

Thank you very much for your thorough revisions to your manuscript based on reviewer comments. I very much appreciate the effort you have put into this and am happy with your responses throughout. I also appreciate the English language edits you have implemented.

I am happy to accept your manuscript in its current form for publication. Thank you very much for your contribution!